# Contact networks have small metric backbones that maintain community structure and are primary transmission subgraphs

**Rion Brattig Correia** [1,2], **Alain Barrat** [3], **Luis M. Rocha** [1,2¤] *

**1** Instituto Gulbenkian de Ciência, Oeiras, Portugal, **2** Department of Systems Science and Industrial Engineering, Center for Social and Biomedical Complexity, Binghamton University, Binghamton New York, United States of America, **3** Aix Marseille Univ, Université de Toulon, CNRS, CPT, Turing Center for Living Systems, Marseille, France

¤ Current address: Department of Systems Science and Industrial Engineering, Binghamton University, Binghamton NY, United States of America
* rocha@binghamton.edu

**Data Availability Statement:** Computations presented in the analysis were performed in Python. Source code is made available on Github at https://github.com/rionbr/socialbackbone. The

## Abstract

The structure of social networks strongly affects how different phenomena spread in human society, from the transmission of information to the propagation of contagious diseases. It is well-known that heterogeneous connectivity strongly favors spread, but a precise characterization of the redundancy present in social networks and its effect on the robustness of transmission is still lacking. This gap is addressed by the metric backbone, a weight- and connectivity-preserving subgraph that is sufficient to compute all shortest paths of weighted graphs. This subgraph is obtained via algebraically-principled axioms and does not require statistical sampling based on null-models. We show that the metric backbones of nine contact networks obtained from proximity sensors in a variety of social contexts are generally very small, 49% of the original graph for one and ranging from about 6% to 20% for the others. This reflects a surprising amount of redundancy and reveals that shortest paths on these networks are very robust to random attacks and failures. We also show that the metric backbone preserves the full distribution of shortest paths of the original contact networks—which must include the shortest inter- and intra-community distances that define any community structure—and is a primary subgraph for epidemic transmission based on pure diffusion processes. This suggests that the organization of social contact networks is based on large amounts of shortest-path redundancy which shapes epidemic spread in human populations. Thus, the metric backbone is an important subgraph with regard to epidemic spread, the robustness of social networks, and any communication dynamics that depend on complex network shortest paths.

distanceclosure Python package used to compute the metric backbones can be found at https://github.com/rionbr/distanceclosure. The SocioPatterns datasets analyzed can be found in http://www.sociopatterns.org/datasets. The Toth et al (2015)[34] and Salathé et al (2010) [36] datasets can be found in their respective publications.

**Funding:** This work was partially funded by National Institutes of Health, National Library of Medicine Program, grant 01LM011945-01 (LMR and RBC), a Fulbright Commission fellowship (LMR), the NSF-NRT grant 1735095 "Interdisciplinary Training in Complex Networks and Systems" (LMR), a CAPES Foundation fellowship, grant 18668127 (RBC), the Fundação para a Ciência e a Tecnologia, grant PTDC-MEC-AND-30221-2017 (RBC), and the Agence Nationale de la Recherche (ANR) project DATAREDUX ANR-19-CE46-0008 (AB). The funders had no role in study design, data collection and analysis, decision to publish, or preparation of the manuscript.

**Competing interests:** The authors declare no competing interests.

## Author summary

It is through social networks that contagious diseases spread in human populations, as best illustrated by the current pandemic and efforts to contain it. Measuring such networks from human contact data typically results in noisy and dense graphs that need to be simplified for effective analysis, without removal of their essential features. Thus, the identification of a primary subgraph that maintains the social interaction structure and likely transmission pathways is of relevance for studying epidemic spreading phenomena as well as devising intervention strategies to hinder spread. Here we propose and study the metric backbone as an optimal subgraph for sparsification of social contact networks in the study of simple spreading dynamics. We demonstrate that it is a unique, algebraically-principled network subgraph that preserves all shortest paths. We also discover that nine contact networks obtained from proximity sensors in a variety of social contexts contain large amounts of redundant interactions that can be removed with very little impact on community structure and epidemic spread. This reveals that epidemic spread on social networks is very robust to random interaction removal. However, extraction of the metric backbone subgraph reveals which interventions—strategic removal of specific social interactions—are likely to result in maximum impediment to epidemic spread.

## 1 Introduction

As the current COVID-19 pandemic illustrates, our social lives and overall public health depend heavily on interactions that scale from the molecular networks of minute pathogens to the complex socio-technical networks of our transportation, health, economy, ecology, and governance systems. Many insights into the organization of such systems come from advances in network science based on the study of patterns of connectivity (network structure), and one of the fields in which network science has led to the most concrete advances is the epidemiology of infectious diseases [1]. In disease propagation, the structure of the contact network plays a crucial role, and networks with heterogeneous connectivity strongly favor spread [2]. Much remains to be understood, however, about how the structure of complex networks affects their dynamics and robustness [1, 3, 4]. For instance, the collection of social behavior data from social media and mobile devices allows us to map the structure of social interaction to understand health and disease in an unprecedented manner [5, 6]. Yet, much remains to be understood about the dynamics of transmission on these networks for us to be able to predict and control specific biomedical phenomena, such as epidemics, affecting society.

Weighted graphs, where every edge is enriched with a positive real number, are often used to capture distance or proximity associations between linked nodes within a set of node variables. This type of network is very useful to infer social dynamics in real-world settings and often used in epidemiology to model disease spread and evaluate disease containment policies at various scales [1, 7–9]. Here we address the link between structure and dynamics by focusing on important *patterns of redundancy* in social networks modeled as weighted graphs.

Redundancy is considered a primary factor in the evolvability of complex systems [10], and recent characterizations of the phenomenon have shown it greatly contributes to their dynamics, controlability, and robustness [11–13]. A full understanding of the interplay between network structure and dynamics requires a study of multivariate dynamics [12] and its redundancy [14]. Often, however, we do not possess enough time-resolved data or computational power to precisely characterize the multivariate dynamics of large networks. In these cases, network structure is very useful for understanding the dynamics of spread and

communication phenomena, which can be inferred from the shortest paths and community structure among the variables. We know complex networks generally contain substantial connectivity redundancy whereby all shortest paths can be computed from the smaller subgraph of the recently introduced *metric backbone* (known in general for any measure of path length as the distance backbone) [13].

Weighted graphs inferred from real-world data can be large and dense, and thus, cumbersome to manipulate [15]. To overcome this problem, several sparsification methods have been proposed to extract network backbones by removing weak, non-significant, or redundant edges. Methods that remove edges based on a distance (or strength) threshold tend to disrupt global network connectivity by altering the distribution of shortest paths and possibly creating disconnected components (islands) [16]. Indeed, even very weak edges may be very important for computing shortest paths [17] and maintaining multiscale organization (including community structure) in complex networks [18]. Other methods to extract backbones are based on comparison with an expected connectivity distribution (i.e., a null model) [15, 18–22] or certain network properties (e.g., degree, betweenness, and effective resistance) [23, 24]—sometimes altering retained edge weights to conform to desired network properties [24]. All methods, ultimately, remove edges (and potentially nodes) based on thresholding edge weights (retaining only the edges with a proximity weight larger than a given value) or comparing to a null-model distribution. Thus, in either case there is an arbitrary parameter that tunes the removal of edges (and nodes).

In contrast, the metric backbone is a parameter-free, algebraically-principled method to obtain a unique (not estimated) subgraph with unaltered weights that fully preserves the shortest paths of the original graph [13]. This means no shortest path is affected by reduction of the original graph to its metric backbone. In particular, even the distance between nodes directly linked by a removed edge is not affected as the shortest path between these nodes is then an indirect path via other nodes (edges not on the backbone break the triangle inequality; see [13] for details).

The metric backbone is typically a subgraph much smaller than the original network across domains ranging from topical spaces of large document corpora [25–27] to the brain connectome and functional networks [13, 28–30]. For instance, the metric backbone of a knowledge graph of more than 3 million concepts extracted from Wikipedia and used for automated fact-checking [31] contains only 2% of the original edges but is sufficient to compute all shortest-paths of the original graph [13]. The small relative size of the metric backbone reveals that network shortest-path robustness to attacks and failures likely stems from surprisingly vast amounts of redundancy [13].

Importantly, other backbone methods end up removing edges that are not redundant for shortest paths. Even the *disparity filter backbone* [18], which has been proposed to preserve the multiscale structure of complex networks, alters the distribution of shortest paths, overall connectivity, and can remove nodes [13]. For instance, the disparity filter backbone of an air traffic network of over 1000 U.S. airports is composed of 24% of the original edges, but it alters the shortest path distribution and removes 23% of all nodes [18] (using $\alpha = 0.2$, which is the p-value threshold for the null model-derived normalized-edge-weight distribution; For $\alpha = 0.05$, the disparity filter backbone is composed of 17% of the original edges but also removes 34% of the original nodes [18]). In contrast, the metric backbone of the same airport traffic network is much smaller, composed of 16% of the original edges, and keeps all node variables with the same connectivity and shortest path length distribution [13]. Moreover, the metric backbone is parameter free (see Section 2) while the disparity filter backbone depends on a significance level parameter in comparison to a null model distribution to remove edges.

Given the desirable properties of the metric backbone, here we use it to better understand how (shortest-path) redundancy shapes the dynamics of epidemic transmission in human populations. We present the metric backbones of nine networks obtained by measuring the contacts between pairs of individuals (using wearable sensors) in a variety of social settings [32–34]. Such networks are relevant in behavioral studies [35] and as input to data-driven numerical simulations of epidemic spread [9, 34, 36]. We already know that the metric backbone preserves all shortest paths in these networks and removes edges truly redundant for this purpose [13]. Here we show that in contact networks from various social contexts, the metric backbones: (a) are much smaller than the original networks, revealing a social organization with much redundancy in connectivity, (b) preserve community structure with all its intra- and inter-community shortest paths, and (c) are primary subgraphs for simple transmission dynamics, revealing that diffusion processes on social contact networks are largely driven by, or canalized via, the metric backbone of the latter. In addition to this novel characterization of the organization of social contact networks, and the role of such organization in transmission dynamics, we discuss how this knowledge is relevant for devising intervention strategies to disrupt spreading phenomena on social networks.

## 2 The metric backbone of contact networks

Social contact networks are built by recording with whom individuals meet and, when possible, for how long. The data we consider (see Section 4) describe close proximity events in populations with a finite temporal resolution, i.e., in successive time windows of approximately 20 seconds [32–34, 37]. These data can be represented as graphs, $R(X)$, in which each node $x_i \in X$ represents a person in a population $X$, and an edge exists between two nodes if the corresponding individuals have been in close contact at least once. The number of time windows in which individuals $x_i$ and $x_j$ were in close proximity is denoted by an entry $r_{ij}$ in the graph adjacency matrix. The diagonal entries of this matrix, $r_{ii}$, denote the total number of time windows in which individual $x_i$ was in a close social interaction with any other individual in the population $X: r_{ii} = \sum_{x_j \in X: j \neq i} r_{ij}$. A normalized transformation of this data, used to quantify the strength of association between individuals, can be obtained via the Jaccard measure [38, 39]:

$$p_{ij} = \frac{r_{ij}}{r_{ii} + r_{jj} - r_{ij}}, \ \forall x_i, x_j \in X, \tag{1}$$

where $p_{ij} \in [0, 1]$ denotes a *proximity* between two individuals, with $p_{ij} = 0$ for individuals $x_i$ and $x_j$ that have no contact, and $p_{ij} = 1$ when they are in contact in every time window measured; naturally, $p_{ii} = 1$. The *proximity graph*, $P(X)$, can be produced by measures other than Jaccard's as long as the proximity strength is symmetrical and proportional to the intensity of the interaction [27]. Two additional forms of normalizing interactions are described in SM, Section A in S1 Text.

Because computing shortest paths requires a measure of length, rather than proximity, we also compute *distance graph*s $D(X)$ obtained via the nonlinear map $\varphi$:

$$d_{ij} = \varphi(p_{ij}) = \frac{1}{p_{ij}} - 1 = \frac{r_{ii} + r_{jj} - 2r_{ij}}{r_{ij}}, \ \forall x_i, x_j \in X, \tag{2}$$

where the resulting distance weights are symmetrical and inversely proportional to the intensity of social interaction, with $d_{ij} = +\infty$ for individuals $x_i$ and $x_j$ who have no contact, $d_{ij} = 0$ when they are always in contact, and $d_{ii} = 0$. Other maps are possible, but without loss of generality $\varphi$ in Eq 2 is the simplest nonlinear isomorphism possible between proximity and distance graphs [27].

The *metric backbone* [13] of a distance graph $D(X)$ is defined as its invariant subgraph $B(X)$ in the computation of the graph's *metric closure* $D^T(X)$. The metric closure is the graph obtained after computing the shortest paths between all pairs of nodes of the distance graph and replacing the original distance edges $d_{ij}$ with the *length of the shortest path* between $x_i$ and $x_j$. Length is computed by summing the edge weights in the (shortest) path via $\delta$ indirect (non-repeating) nodes, where $\delta$ is no larger than the diameter of the graph, thus, $d_{ij}^T = \ell_{ij} = d_{ik_1} + d_{k_1 k_2} + \ldots + d_{k_\delta j}$. The edge weights of the metric backbone graph are given by:

$$b_{ij} = \begin{cases} d_{ij}, & \text{if } d_{ij} = d_{ij}^T \\ +\infty, & \text{if } d_{ij} > d_{ij}^T \end{cases}, \forall x_i, x_j \in X, \tag{3}$$

where $b_{ij} = +\infty$ means that there is no direct edge between $x_i$ and $x_j$ in the distance backbone graph. The metric closure is one of infinite possible distance closures that are isomorphic to transitive closures in generalized probabilistic or fuzzy metric spaces. In this case, the topological space is defined by the algebraic structure $([0, +\infty], \min, +)$, where $([0, +\infty], \min)$ and $([0, +\infty], +)$ are monoids that enforce shortest paths with lengths computed by summing edge weights. There are many other ways to compute length besides summing edges [27] that lead to many other meaningful distance backbones [13]. However, the metric closure, or the *All Pairs Shortest Paths* (APSP) problem, with computational complexity in the range $\mathcal{O}(|X|^3)$ [27, 40], is the most common in network science and typically computed via Dijkstra's algorithm [41].

The edge weights of $D(X)$ that do not change after computation of the metric closure $D^T(X)$ are called *metric* because they obey the *triangle inequality*:

$$d_{ij} \leq d_{ik}^T + d_{kj}^T, \forall x_i, x_j, x_k \in X. \tag{4}$$

The edge weights that become smaller with the metric closure break the triangle inequality in $D(X)$ and are not included on the backbone. When an edge $d_{ij}$ of $D(X)$ breaks the triangle inequality, it means that the *length* of at least one indirect path between $x_i$ and $x_j$ is shorter than the direct distance: $\ell_{ij} = d_{ij}^T < d_{ij}$. These are known as *semi-metric edges* [25] and do not contribute to any shortest path [27]. Thus, metric edges alone define the backbone and are sufficient to compute the closure, as shown in [13]. Notice that this construction is based on edge, not path, properties. If every edge on equivalent shortest paths between nodes $x_i$ and $x_j$ happen to be metric, they all remain in the metric backbone, along with the possible paths they may form. This means that several alternative paths of the same length can exist between any two nodes in the backbone, which is quite distinct from path-based graph reductions such as Minimum Spanning Trees [13].

The *proportion of semi-metric edges* is, therefore, the proportion of edges in graph $D(X)$ that are not necessary to compute shortest-paths. This measure of *edge redundancy* is given by:

$$\sigma(D) = \frac{|\{d_{ij} : d_{ij} > d_{ij}^T\}|}{|\{d_{ij}\}|}, \forall x_i, x_j \in X : i > j. \tag{5}$$

Similarly, the *proportion of metric edges* in graph $D(X)$ is the *relative size of its metric backbone B(X)*:

$$\tau(D) = \frac{|\{d_{ij} : d_{ij} = d_{ij}^T\}|}{|\{d_{ij}\}|} = \frac{|\{b_{ij}\}|}{|\{d_{ij}\}|}, \forall x_i, x_j \in X : i > j. \tag{6}$$

It follows that $\tau = 1 - \sigma$. Because distance graphs are symmetric ($d_{ij} = d_{ji}$) and edges are nondirected, in Eqs 5 and 6 we count each edge only once and do not tally reflexive edges, $d_{ii}$. This means we tally only the lower diagonal entries of the adjacency matrix: $d_{ij} : i > j$.

The metric closure, computation of all pairs shortest paths (APSP), induces a topological *distortion* [27] of the original graph obtained from the multivariate associations observed in the social contact data, whereby semi-metric edges are made to conform to the triangle inequality, Eq 4. However, only the semi-metric edges get distorted; the metric edges and the metric backbone they compose remain invariant in the APSP. Therefore, $\sigma$ also denotes the proportion of edges topologically distorted by the metric closure, whereas $\tau$ denotes the proportion of edges topologically invariant under the metric closure. A measure of *semi-metric edge distortion* between nodes $x_i$ and $x_j$ is then obtained via the ratio of the direct distance over the shortest indirect path length.:

$$s_{ij} = \frac{d_{ij}}{d_{ij}^T}, \forall x_i, x_j \in X : i \neq j. \tag{7}$$

If an edge $d_{ij}$ is metric, $s_{ij} = 1$, meaning there is no distortion. If an edge $d_{ij}$ is semi-metric, $s_{ij} > 1$, meaning there is distortion; the larger the value over 1, the more the edge breaks the triangle inequality, and thus, the more distorted it is in the metric closure. While semi-metric edges are redundant for shortest paths and have null betweenness centrality, their distortion (the distribution of $s$) varies widely. This in turn affects the robustness of shortest paths to edge removals [13] (see Fig 5 and Section 3.2).

It is important to note that the nonlinear map $\varphi$ (Eq 2) has been shown to establish an isomorphism between proximity graphs $P(X)$ (Eq 1) and distance graphs $D(X)$ (Eq 2) [27]. Therefore, the distance backbone $B(X)$ exists as a subgraph of both $D(X)$ and $P(X)$, with all associated properties and formulae (Eqs 3–7). Strictly, the distance backbone of a proximity graph, $T(X)$, is known as a *transitive backbone* (a kind of transitive reduction) [13], and is the counterpart of $B(X)$ in proximity space via isomorphism $\varphi$: $t_{ij} = \varphi^{-1}(b_{ij}) = 1/(b_{ij} + 1)$. In other words, $t_{ij} = p_{ij}$ if there is a finite edge weight between $x_i$ and $x_j$ in $B(X)$, and 0 otherwise. For simplicity, hereafter we refer to the distance backbone as $B(X)$ but use its distance ($b_{ij}$) or proximity ($t_{ij}$) edge weights depending on the edge interpretation one needs, which we indicate throughout; e.g. distance weights are used for computing shortest paths (using the Dijkstra algorithm), and proximity weights are used as transition probabilities in epidemic spread experiments or as connection strengths in community detection.

## 3 Results

We have computed the metric backbone for nine different contact networks from a variety of social contexts, from an elementary school in Utah (USA) to an art exhibit in Dublin (Ireland). As shown in Table 1, these networks vary widely in the number of nodes and edges, yet the metric backbone typically comprises a small proportion of edges.

Our metric backbone analysis is exemplified by the contact network collected by the Socio-Patterns collaboration [37] over a period of 4 days in 2013 in a High School (Fr-HS) in Marseilles, France [46]. A similar analysis of all other networks is provided in Supplemental Materials (SM). The Fr-HS contact network was built from 188,508 contact records among $|X| = 327$ students enrolled in distinct *préparatoires* classes, which are designed for college-bound students in the last two years of their high school studies. This student body had been largely separated from other high school students (i.e., different building and lunch), forming an almost closed population with few contacts with the outside world, at least during school days. Nine classes were organized in four different specializations: "MP" focus on mathematics

**Table 1. Distance graphs (D) of the social contact networks analyzed, with respective references.** $|X|$: number of nodes; $(|d_{i>j}|)$: number of finite distance edges; $\tau(D)$ relative size of the metric backbone; $\sigma(D)$: edge redundancy. Values of $\tau$, and $\sigma$ are shown as percentages (%). For the Exhibit networks, values of $|X|$, $(|d_{i>j}|)$, $\tau$ and $\sigma$ denote the mean ± standard deviation of the networks computed for each of the 69 days for which data were gathered. See Section 4.1 for additional details and a description of the networks.

| Network (D) | Location | Social context | $|X|$ | $|d_{i>j}|$ | $\tau(D)$ | $\sigma(D)$ |
|---|---|---|---|---|---|---|
| Fr-Ho [42] | Lyon, France | Hospital | 75 | 1,139 | 19.05 | 80.95 |
| It-SC [43] | Turin, Italy | Scientific Conference | 113 | 2,196 | 14.03 | 85.97 |
| Ir-Ex [43] | Dublin, Ireland | Exhibit | 159±63 | 645±468 | 48.44±9.27. | 51.56±9.27. |
| Fr-Wo [44] | Paris, France | Workplace | 232 | 4,274 | 17.43 | 82.57 |
| Fr-PS [45] | Lyon, France | Primary School | 242 | 8,317 | 9.5 | 90.5 |
| Fr-HS [46] | Marseille, France | High School | 327 | 5,818 | 10.36 | 89.64 |
| US-ES [34] | Utah, USA | Elementary School | 339 | 16,546 | 6.82 | 93.18 |
| US-MS [34] | Utah, USA | Middle School | 591 | 56,867 | 6.19 | 93.81 |
| US-HS [33] | USA | High School | 788 | 118,291 | 7.84 | 92.16 |

and physics (3 classes), "PC" on physics and chemistry (2 classes), "PSI" on engineering (1 class), and "BIO" on biology (3 classes). The total student participation in the study was 86.3% [46].

The original contact network and its metric backbone have been obtained for the Fr-HS data via Eqs 1 to 3 in Section 2 and are depicted in Fig 1, with other processing details described in Section 4.1.

## 3.1 Metric backbone and community structure

As shown in Table 1, the Fr-HS metric backbone comprises only $\tau \approx 10\%$ of the edges of the original network. Interestingly, in addition to preserving all shortest paths by design [13], the metric backbone also preserves and highlights the community structure of the original social network. This is clearly seen in Fig 1A–1C when we compare its center (B) and right (C) panels. For the metric backbone depicted in Fig 1B the nodes are placed according to the computation of the `ForceAtlas2` algorithm [47] (in `Gephi` [48]) using all the almost six thousand edges (Table 1) of the original network (shown in Fig 1A). In contrast, in Fig 1C the nodes are placed after recomputing the same algorithm using only the 603 edges of the backbone subgraph. It is clear that the community structure remains largely unaltered with respect to the Fig 1B panel.

Similar results have been observed for all other networks in Table 1 (see Section C in S1 Text), with the interesting exception of the American high school (US-HS) contact network. In this case, the original graph is devoid of any obvious social structure. After edge removal, however, four distinct communities that likely correspond to the four grades typical of American high school education can clearly be seen (see Fig 1D–1F). For this larger network ($|X| = 788$), removing the $\sigma \approx 92\%$ of edges that are redundant for shortest paths reveals a clearer community structure. Moreover, while the metric backbone subgraph is comprised of only $\tau \approx 8\%$ of the edges, it preserves all the original shortest paths, be them inter- or intra-community. Therefore, the metric backbone must preserve the multiscale *distance* structure (or topology) of complex networks.

It is useful to quantify how well the metric backbone preserves or highlights the social organization of contact networks beyond visual inspection. Indeed, while visual representation of networks is an important part of social network analysis, especially for small- or mid-size networks, it becomes increasingly difficult as the number of nodes and edges grows. Therefore, we have developed and used a set of measures to compare the community structure of each

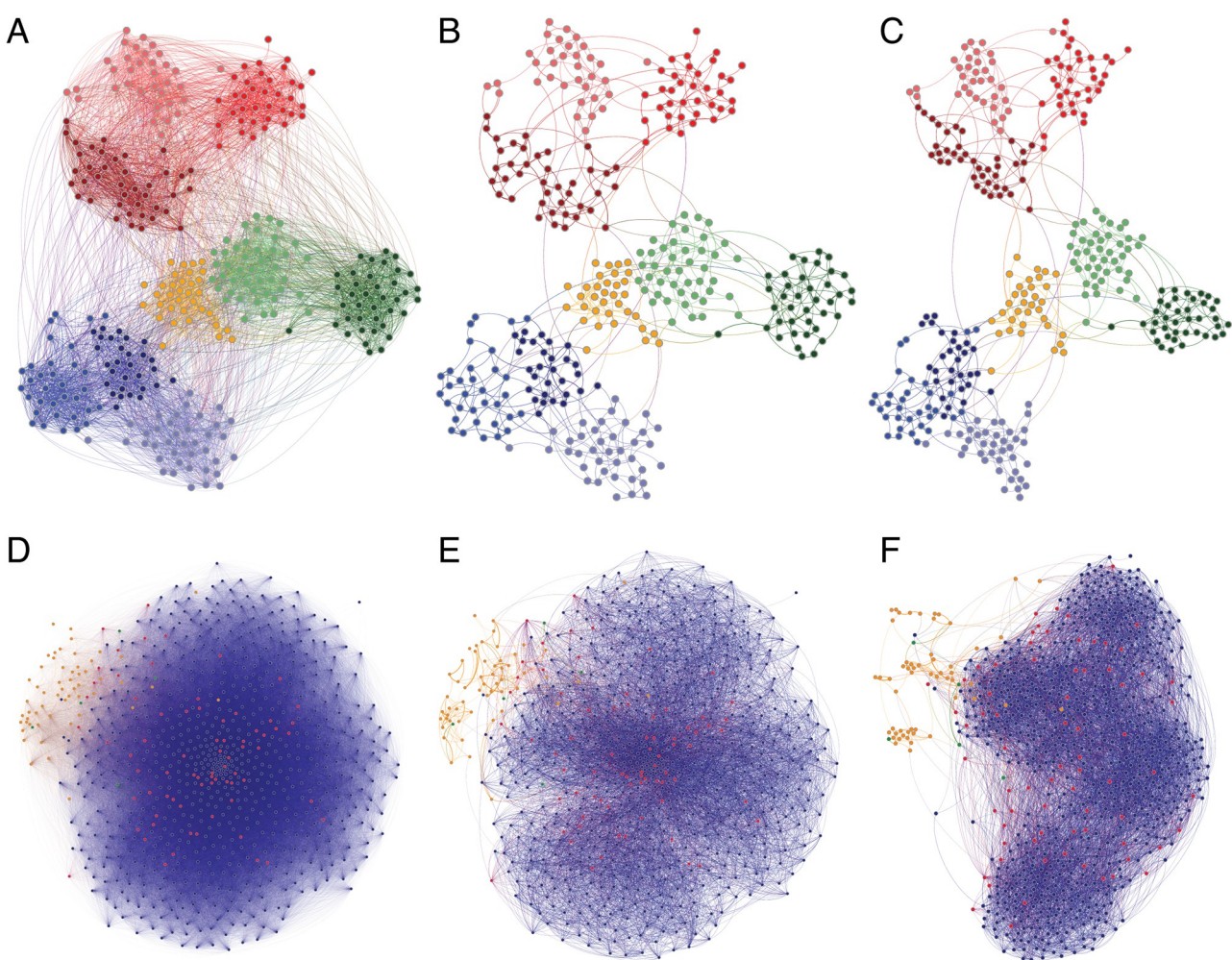

**Fig 1. French high school (Fr-HS) and American high school (US-HS) contact networks.** (**A-C**) French high school (Fr-HS) contact network of $|X|$ = 327 students. Colors represent the four student specializations: "MP" in blue, "PC" in green, "PSI" in orange, and "BIO" in red; lighter or darker colors separate the distinct classes within each specialization. (**D-F**) American high school (US-HS) contact network of $|X|$ = 788 students, staff, and teachers. Colors represent students in blue, teachers in red, and staff in orange. Other (unspecified) individuals are shown in green. No student class metadata is available. (**A, D**) Original networks with node layout computed by `ForceAtlas2` algorithm [47], using all proximity weights, $P(X)$. (**B, E**) Metric backbone subgraphs rendered with the same node layout as in the respective networks in (**A**) and (**D**). (**C, F**) Metric backbone subgraphs with node layout recomputed by `ForceAtlas2` using only backbone (proximity) weights. Plotted with `Gephi` [48].

metric backbone subgraph with its original graph (see Section 4.2 for details). Because all measures support the same conclusions, in Table 2 we present results only for the *bidirectional similarity measure*, $y_{AB}$, as defined by Eq 8 (Section 4.2), applied to the Fr-HS social contact network. Results for all measures and all other networks are provided in Section C in S1 Text.

The Fr-HS dataset includes $m$ = 9 classes in the metadata (see Fig 1A–1C), while the `Louvain` community detection algorithm [49] identifies $m$ = 10 communities in the original graph—which has an additional very small community splitting from one of the MP classes, as shown in Fig 2A. In other words, the community structure of the original graph obtained from all the measured social contact data, as captured by `Louvain`, is very similar to the metalabel communities. Indeed, $y_{AB}$ = 0.88 measures a high amount of bidirectional similarity between the two module partitions—the larger the value of $y_{AB} \in [0, 1]$, the more similar the community structure between the two graphs defined on the same set of nodes, see Section 4.2. This

**Table 2. SocioPatterns Fr-HS contact network and its community structure detected by the Louvain algorithm [49] for the original (proximity) network and several of its subgraphs.** Top rows ($m$) show the number of distinct metalabels and the number of communities detected, while bottom rows show the bidirectional modularity similarity, $y_{AB}$ (Eq 8 in Section 4.2). Columns show values for original graph, metric backbone subgraph, and same-size threshold and random subgraphs. Mean and standard deviation shown for 100 random subgraphs.

| | | Original | Metric | Threshold | Random |
|---|---|---|---|---|---|
| $m$ | Metalabels | 9 | - | - | - |
| | Louvain | 10 | 17 | 27 | 47.19±3.79 |
| $y_{AB}$ | Metalabels | 0.88 | 0.68 | 0.55 | 0.38±0.02 |
| | Original | - | 0.74 | 0.61 | 0.37±0.02 |

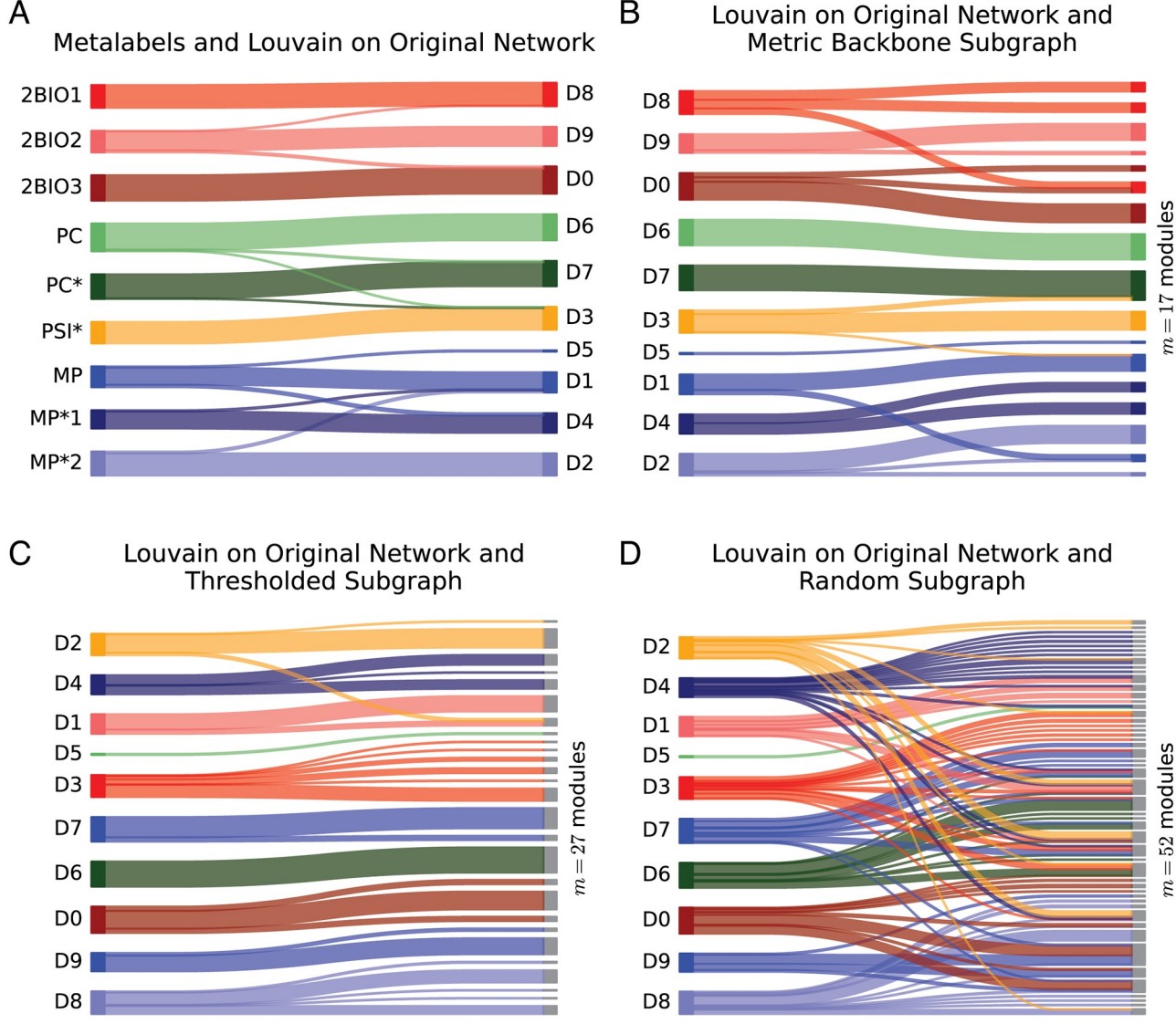

**Fig 2. Community structure comparison for the French high school (Fr-HS) dataset using Sankey plots.** (**A**) Comparison of metalabels—student attribution to their classes—with modules detected on the original graph. (**B**) Comparison of the modules detected on the original graph with those detected on its metric backbone subgraph. (**C**) Comparison of the modules detected on the original graph with those detected on the threshold subgraph of same size as the metric backbone. (**D**) Comparison of the modules detected on the original graph with those detected on a random subgraph of the same size as the metric backbone. Metalabel module colors assigned to student specialization and classes as in Fig 1A–1C. Modules computed using the Louvain algorithm [49].

confirms that most social contacts of students occur within their assigned classes, with only few students who interact more with students from other classes/modules—though almost always within the same specializations, as shown in Fig 2A.

However, we do not have access to a "gold standard," such as an independent survey of the observed cohorts, corresponding to the "true" community structure of the contact networks we analyze. We only have access to either the metalabels provided in the data (e.g., the classes each student of the Fr-HS network is enrolled in), or the social organization uncovered by community detection algorithms, such as the Louvain algorithm applied to the original network. It is important to note that neither is guaranteed to perfectly characterize the true social organization. While the metalabels denote organizational roles (e.g., classes and professions), individuals may contact members of other (metalabel) groups more than those within their own group due to personal relationships outside of their organizational roles. Additionally, the original network may contain spurious contact edges, especially if the sensor parameters defining when a contact is detected do not correspond to close proximity, or if contacts with very short duration are registered. The social organization of the original network as captured by community detection algorithms may thus be confounded by many contact edges that do not in reality denote a strong social link. Indeed, the US-HS network discussed above is one such case, as shown in Fig 1D–1F. Therefore, to provide evidence that the metric backbone presents a good compromise between preserving a substantially correct picture of the organization of contact networks while implementing sizeable and effective network sparsification, we compare its community structure with same-size subgraphs obtained by thresholding or randomly removing edges— see Section 4.3 for details on their generation. Note that we do not compare community structure to other backbone methods, such as the disparity filter backbone [18], because they can remove nodes (see Section 1), which would result in a different universe $X$ to partition into communities. Additionally, distance backbones such as the metric backbone are parameter-free, so we compare its community structure to threshold and random graphs obtained by removing exactly the same number of edges as removed in forming the metric backbone ($\sigma(D).|d_{i>j}|$) in a parameter-free manner. A comparison and discussion of the advantages of distance backbones in regard to alternative backbone subgraphs is already available [13].

In the case of the Fr-HS network, it is clear from Table 2 that the metric backbone preserves the community structure of the original graph found by the Louvain community detection algorithm, as well as that of its metalabel partition, better than the same-size threshold and random subgraphs. The measure of bidirectional modularity similarity of the metric backbone ($y_{AB} = 0.74$), with only about 10% of the edges, shows that it preserves the Louvain-community structure of the original graph much better than the same-size threshold subgraph ($y_{AB} = 0.61$) and than a sample of 100 same-size random subgraphs ($y_{AB} = 0.37 \pm .02$). The difference can also be visualized as in Fig 2, where, by comparing the right-hand side of panels B, C and D, it is clear that the metric backbone breaks into fewer modules ($m = 17$, panel B) than do the threshold ($m = 27$, panel C) and random ($m = 47.19 \pm 3.79$, panel D) subgraphs. In the case of the threshold graph this is likely due to deletion of (weak) bridges within and between modules that are nonetheless required to preserve shortest paths [13]. It is also clear that random deletion not only breaks community structure more, it also mixes up the social organization, with students from different specializations being grouped together in small modules more than as observed with the metric backbone or threshold subgraph. While there is some variation across different social networks, as detailed in SM, the metric backbone typically preserves the community structure of the original graph better than same-size threshold and random backbones for the various similarity measures defined in Section 4.2.

A similar behavior occurs with the metalabel partition, which is also better captured by the metric backbone ($y_{AB} = 0.68$) than the same-size threshold subgraph ($y_{AB} = 0.55$) and the

sample of 100 same-size random subgraphs ($y_{AB} = 0.38 \pm .02$). However, as mentioned above, it is not clear that metalabel partitions capture the true social organization, especially in several of the networks analyzed in SI where metalabels are not always expected to form social contact modules in the observed contexts (e.g., teachers in classroom settings or in-patients in a hospital). Still, in school settings like that of the Fr-HS network, it is clear that threshold and random subgraphs break the metalabel partition into many more `Louvain`-communities than does the metric backbone, and the random subgraphs mix the metalabel partition much more. Interestingly, with the French Primary School network (Fr-PS) we observe that the `Louvain`-communities of the metric backbone can be closer to the metalabel partition than the `Louvain`-communities of the original graph (see Section C.1 in S1 Text), which suggests that the metric backbone can in some cases filter out noisy or redundant contact data, as was also observed in the US-HS network (Fig 1D–1F).

In addition to the various measures of module similarity and `Louvain`-communities, we have additionally used the `Infomap` community detection algorithm [50] and stochastic block models (SBM) [51] to study the ability of the metric backbone to preserve the community structure of original networks (including their metalabels) in comparison to same-size threshold and random subgraphs. For the analyzed social contact networks `Infomap` leads to a much more granular community structure, typically breaking it into many small modules. Indeed, the SBM simulations show that `Infomap` is not able to recognize well the known original communities of full networks when connectivity is relatively low, even prior to any sparsification (see Section B in S1 Text, especially Table C). Thus, even though the results generally support the same conclusions drawn with `Louvain`-communities, we include the `Infomap` results only in the SM, for the SBM simulations (Section B in S1 Text) and for every network studied (Section C in S1 Text).

We used SBM to generate ensembles of artificial networks for which we can control how much group metalabels match the underlying community structure—something we cannot control in the real-world contact networks as discussed above. The simulations consider two distinct scenarios for low and high connectivity, as well as modules with and without hierarchical structure. First, for both low and high connectivity, the metric backbone preserves well the community structure of the original network (see Section B in S1 Text for details). Even in the most adverse scenario of low connectivity, this can be clearly observed by inspecting the adjacency matrix and the network visualization in Fig 3. In particular, it is clear that the backbone preserves the original community structure when its subgraph is independently plotted with the same force layout algorithm as the original, full network. Furthermore, the community structure detected on the metric backbone subgraphs is systematically more similar to the one detected on the full generated networks than the community structures of threshold and random subgraphs of the same size. This is the case for both the high and low connectivity SBM, for Louvain and Infomap community structure detection methods, and across most measures of modularity similarity (see Tables B and C in S1 Text).

Altogether, the analyses of community detection, module similarity measures, visualization, and SBM simulations support the assertion that the metric backbone presents a good compromise between preserving a substantially correct picture of the organization of contact networks while implementing sizeable and effective network sparsification.

## 3.2 Epidemic spreading on the metric backbone

Networks are the supporting structure of various types of dynamical processes, ranging from the synchronization of oscillators to the spread of information or infectious diseases [2]. How these processes are altered when they occur on a subgraph, with respect to how they would

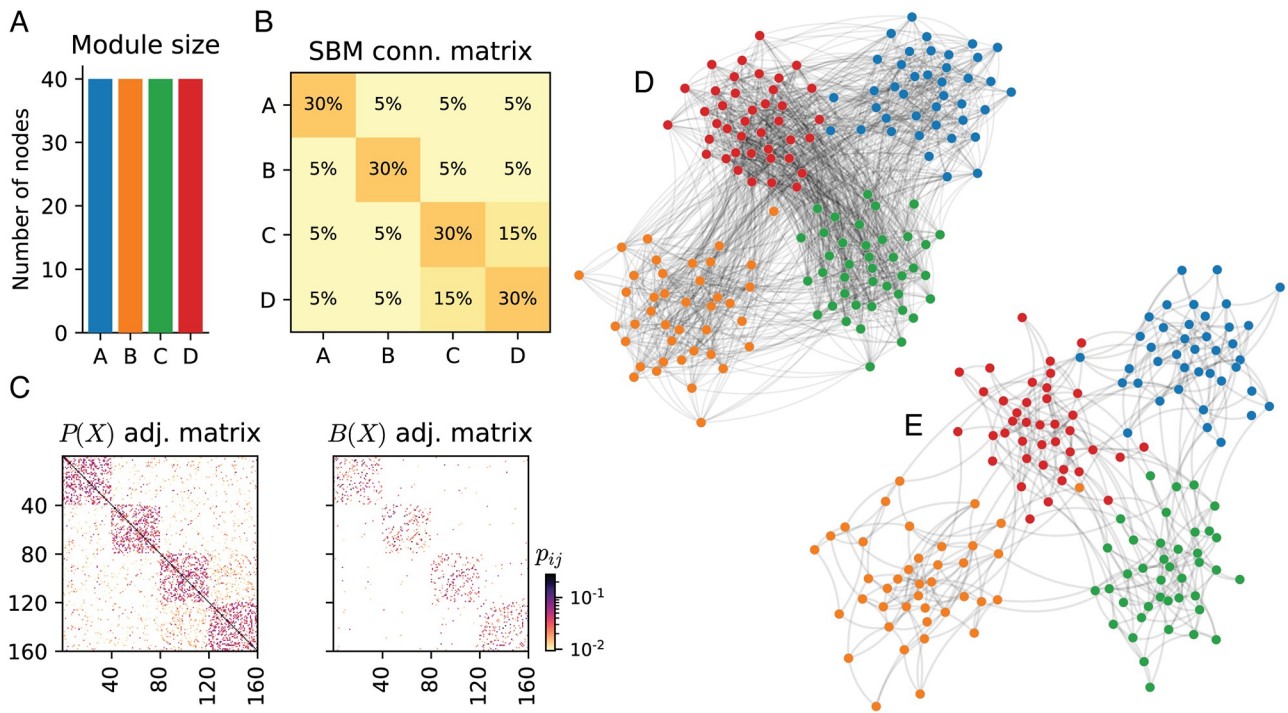

**Fig 3. Synthetic network with connectivity generated by a Stochastic Block Model (SBM) and edge weights sampled from the SocioPatterns French Primary School (Fr-PS) contact network.** (**A**). Synthetic module sizes. (**B**). SBM generator matrix for low connectivity case. Note the hierarchical structure of modules C and D. (**C**). The adjacency matrix of the original graph and the metric backbone. (**D**). Force layout visualization of the original (proximity) graph as computed by the NetworkX python package [52]. Node positioning obtained after 100 iterations. (**E**). Force layout visualization of the metric backbone subgraph. Node positioning seeded from the original graph (panel **C**) and obtained after 100 iterations. See Sections A and 4 in S1 Text for details about synthetic network generation.

unfold on the whole network, is a natural question when devising a filtering strategy that retains only a subgraph of the original network [24, 53]. We thus explore this issue in the case of the metric backbone.

As many processes are based on diffusion phenomena, we focus on simple and paradigmatic contagion processes typically used to simulate the spread of infectious diseases in a population. Moreover, because our focus is on how a spreading process occurring on a whole network differs from that on the metric backbone (i.e., we are not interested in recovering dynamics) we consider the simplest such model, the *Susceptible-Infected* (SI) model. In this model, each node can be in only one of two states: S (susceptible) or I (infected and infectious). A susceptible node, $x_j$, in contact with an infectious one, $x_i$, along an edge of (proximity) weight $p_{ij}$ can become infectious at rate $\beta.p_{ij}$, i.e., with probability $\beta.p_{ij}.dt$ in each time interval $dt$. In this section we use proximity weights since the infection probability increases with proximity—equivalently, it decreases with the isomorphic distance since $p_{ij} = 1/(d_{ij} + 1)$ and the distance backbone, as shown in section 2, exists equally in distance graphs and their isomorphic proximity graphs. Note further that, on a static network, $\beta$ is simply a global parameter that sets the timescale, so the results are independent from its value as shown below. Once infected, nodes remain in that state.

Each simulation starts with an infectious seed node chosen at random and ends when no further contagion event is possible (either because all nodes are in state I or because the remaining S nodes have no link with the infectious nodes). In order to characterise the spread, we do not focus on the final size (number of nodes reached by the spread, which is always

close to the total number of nodes as they do not recover) but rather on the velocity of the spread [54]. More specifically, we compute the time needed by the process to reach either half of the nodes ($t_{1/2}$) or all the nodes ($t_1$) [54–57], both on the original network and on its metric backbone subgraph, using their proximity weights in the SI spreading process.

In addition to comparing spreading times on the original network with those using the metric backbone, as in the Section 3.1 analysis, we also consider two baselines: subgraphs obtained either by thresholding the weights or by randomly removing edges As noted in Section 3.1, we do not compare to other backbone methods because they can remove nodes in addition to edges, change the original weights, or introduce parameters that would complicate the transmission study pursued. A discussion of the advantages of distance backbones in regard to alternative backbone subgraphs is available [13].

For a fair comparison among metric backbone, threshold, and random subgraphs, we need to contrast the various subgraphs using the same number of edges. However, in many cases the threshold and random subgraphs are composed of several disconnected components though they have the same number of edges as the metric backbone, as only the metric backbone guarantees that connectivity and shortest path distribution are preserved [13]. Therefore, we have devised the following procedure: starting from the whole network, we remove a fraction of the semi-metric edges randomly in order to obtain a subgraph composed of (i) the metric backbone (ii) plus a percentage $\chi$ of semi-metric edges (similar to the procedure in [53]).

For this study, for every network in Table 1, we have performed $n_r = 10$ runs of the SI model starting from a random node seed and computed $t_{1/2}$ and $t_1$ averaged over these runs and over 100 network realizations of the set of randomly selected semi-metric edges that are not removed. Thus, for each value of $\chi$, 1000 simulations are computed. We then built threshold and random subgraphs with exactly the same number of edges as the metric backbone, and we performed exactly the same procedure in each case to obtain a series of intermediate subgraphs between the original graph and the final threshold or random subgraphs of the same size as the metric backbone, i.e., we randomly removed a fraction $1 - \chi$ of the edges of the original graph that did not belong to the threshold or to the random subgraph, respectively.

Fig 4 depicts $t_{1/2}$ and $t_1$, normalized by their values computed on the original graph, as a function of $\chi$ in the case of the French High School (Fr-HS) network. The rightmost point corresponds to the whole original network, $D(X)$, and the leftmost to the metric backbone, $B(X)$, or same-size threshold and random subgraphs. The simulation results for the metric backbone are depicted by the red circle, full line. The corresponding results for same-size threshold and random subgraphs are depicted by the blue triangle, dash-dotted and green triangle, dotted lines, respectively. In all cases, as edges are removed from the original network, the time to infection or half-infection increases as fewer paths are available to transmit the disease: any filtering method leads to an overestimation of infection times. However, it is quite clear that the infection on the metric backbone propagates faster and remains much closer to the infection times observed for the full original network than the infection times observed on the baseline subgraphs. This is especially striking for the time to full infection $t_1$, but also observed for time to half infection $t_{1/2}$: the ratio between the times measured on the original network and on the metric backbone reaches $\approx 1.6$ for $t_1$ and $\approx 2.3$ for $t_{1/2}$, but much larger ratios are reached with the other two baseline subgraphs. Similar results are observed for every network in Table 1, as detailed in SM. Note that we report results for a given value of $\beta$, since it only fixes a global timescale for the spread. Indeed, we have verified that the normalized times reported do not depend on $\beta$. We show in the Fig H in S1 Text the non-normalized infection times for another value of $\beta$.

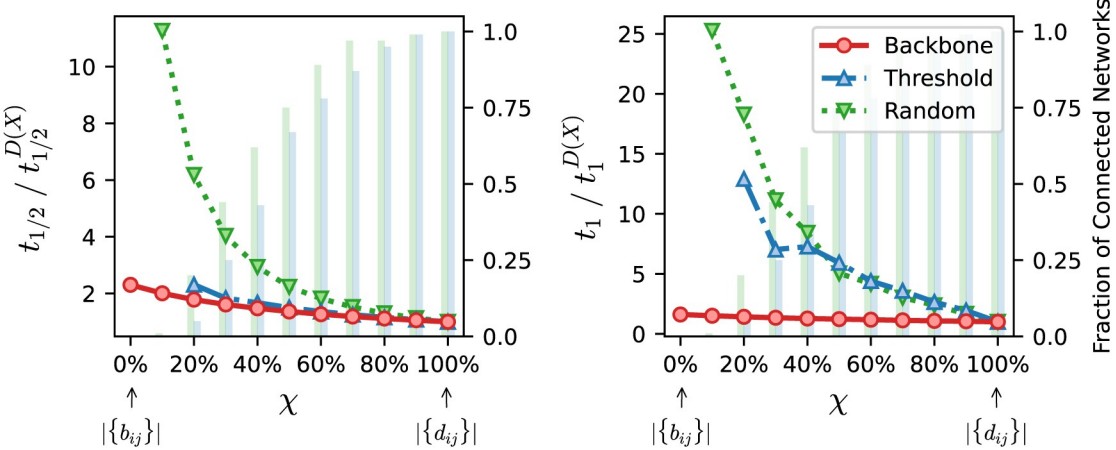

**Fig 4. Normalized time to infection using the metric backbone, threshold, or random subgraphs of the French High School (Fr-HS) network.** The horizontal axis denotes $\chi$, a parameter to sweep the proportion of edges of the original network that are included in the subgraphs analyzed. When $\chi = 0\%$ (leftmost value on axis) we have the metric backbone subgraph, or threshold and random subgraphs with the same number of edges as the metric backbone (i.e. $|\{b_{ij}\}| = \tau(D).|\{d_{ij}\}|$, per Eq (6)). As $\chi$ increases, edges from the original network that are not on the backbone or same-size threshold and random subgraphs, are progressively added until the original network itself is reached at $\chi = 100\%$ (see text for more details). (**Left**) Time for half of the population to be infected, $t_{1/2}$, normalized by the results obtained using the entire original network, $t_{1/2}^{D(X)}$. (**Right**) Time for all nodes in the network to be infected, $t_1$, normalized by the results obtained using the entire original network, $t_1^{D(X)}$. Spreading times for every curve are averaged over $n_r = 10$ runs of the SI model starting from a random seed node and 100 network realizations obtained via random edge removals for each value of $\chi$. The green and blue bars, quantified against the right vertical axis in each panel, denote the fraction of networks in threshold and random baseline ensembles that are connected for a given $\chi$. Disconnected networks are discarded to compute the spreading times. For the simulations shown, the spreading parameter was set as $\beta = 0.9/p_{max}$ where $p_{max}$ is the largest proximity weight of the original network.

Another noteworthy observation ensues from the simulations conducted on all networks. As $\chi$ was decreased we frequently obtained disconnected threshold and random subgraphs, for which $t_1$ is by definition infinite as not all nodes could be reached, and $t_{1/2}$ is finite only if the seed node existed in a connected component that contained at least half of the nodes. The fraction of connected networks for both baselines is indicated by blue and green bars in Fig 4. This fraction becomes rather small well before we reach the size of the metric backbone ($\chi = 0$). Indeed, for $\chi = 20\%$, only 5% and 20% of connected networks exist in the threshold and random baseline ensembles, respectively. Naturally, the values of $t_1$ and $t_{1/2}$ shown are only computed for connected networks, thus simulations for subgraphs of the same size as the metric backbone ($|B(X)|$) were often not possible for both threshold and random baseline ensembles; in the case of the threshold baseline, none of the subgraphs with $\chi = 10\%$ were connected. This means that we cannot find threshold and random subgraphs of the same size as the metric backbone that can sustain propagation to the whole (or even half) of the original network as $\chi$ gets small. In contrast, time to full (or half) infection is always finite in the case of the backbone—and any intermediate case between it and the full network as $\chi$ varies—as it preserves the connectivity (and shortest-path distribution) of the original network.

Altogether the SI simulations on every network in Table 1 show that the metric backbone is a primary subgraph for simple transmission dynamics. Furthermore, considering that the threshold subgraph is built by removing the weaker edges (low proximity or large distance), it follows that there are weak links in the metric backbone that are removed from the network in the threshold subgraph but which are essential for transmission. These weak edges are potentially good candidates for interventions aiming at reducing transmission while minimizing reduction of social contact.

## Discussion

It is only in weighted graphs—where weights discriminate and characterize the degree of association between nodes as proximity or its isomorphic distance—that the concept of metric backbone is meaningful because the backbone of non-weighted (regular) graphs is the graph itself. In weighted graphs the metric backbone (and its generalized distance backbone [13]) is a very useful construct because, unlike other graph reduction techniques, the backbone graph $B(X)$ is guaranteed to preserve all nodes, bridges, original weight values, connectivity, and distribution of shortest paths of the original graph $D(X)$, and is defined in a non-parametric manner from simple algebraic principles. Thus, the metric backbone of a distance graph is unique, typically small, and provides a principled graph reduction technique that does not alter edge weights while preserving all shortest paths.

We have analyzed the metric backbones of nine contact networks collected in a variety of social settings and recorded based on the interactions between pairs of individuals via wearable sensors. It was already known that the metric backbone, by design, preserves network connectivity and all shortest paths after removal of all redundant edges for that purpose [13]. Here we have shown that the metric backbones of social contact networks are also much smaller than their associated original networks (large redundancy in shortest-path calculation), as previously observed in biological, technological, and knowledge networks [13]. Built-in redundancy is a hallmark of complex systems, allowing them natural protection against perturbations [58, 59]. In the social networks analyzed here, the small backbones reveal this in that their distribution of shortest paths is very robust to random removal of edges. Specifically, the proportion of semi-metric edges ($\sigma$) ranges from 52 to 94%, with eight of the nine networks observing $\sigma >$ 80% redundancy. This means that all shortest paths can be computed in these eight networks with fewer than $\tau = 20\%$ of the edges. Indeed, five networks have metric backbones composed of $\tau \leq 10\%$ of their original edges (see Table 1). This suggests that the organization of social contact networks is based on large amounts of shortest-path redundancy.

The only network that has a fairly large metric backbone ($\sigma = 48 \pm 9\%$) pertains to a context where we do not expect a typical social organization: an exhibit in the Dublin Science Gallery (Ir-Ex). In a museum setting, most visitors pass by alone or in small disconnected groups, thus a large fraction of (weak) edges are required to preserve shortest-path connectivity. Indeed, as shown in Fig 5, the distribution of semi-metric distortion reveals a social organization of almost random encounters. The values of semi-metric distortion are quite low ($\tilde{s}_{ij} \approx 2$) with a small variation range (see also Fig S in S1 Text). This means that the distance graph is very close to being entirely metric, as previously observed in random fluctuation phenomena that occurs in an physical (Euclidean) space [13]. In such cases, the backbone is expected to comprise 50% of the graph, but the (near 50%) semi-metric edges are almost all very close to metric (low distortion).

Because shortest paths reveal the strength of social connectedness and the likelihood of transmission phenomena on these networks, we have focused on analyzing how both are captured by the metric backbone. Community structure refers to the identification of subsets of nodes that are more connected among themselves, than to nodes in other subsets. In weighted graphs, this means subsets of nodes that are nearer one another while farther away from other subsets) Since the distance backbone preserves all shortest paths (isomorphically in both proximity or distance graphs, as discussed in Section 2), it must preserve all intra- and inter-community shortest distances between every pair of nodes. This is an important theoretical result about the metric backbone regarding community structure which we also explored experimentally. Indeed, the metric backbone clearly helps in the identification and visualization of the social community structure of the network, especially when the original graph is too dense

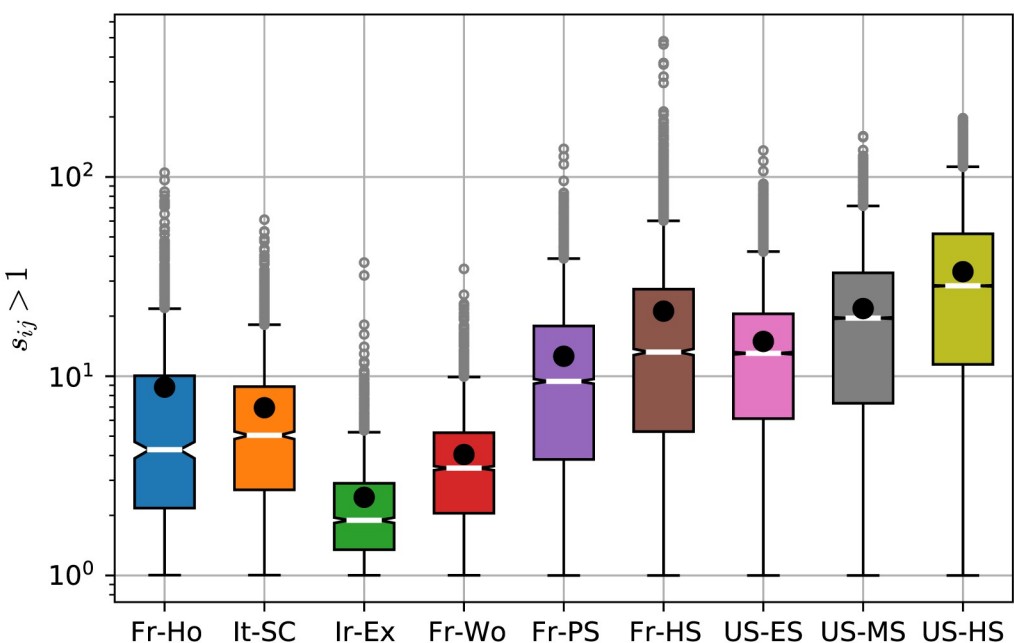

**Fig 5. Distribution of semi-metric edge distortion for contact networks.** Box plots of the distributions of semi-metric distortion for semi-metric edges ($s_{ij} > 1$) for all contact networks in Table 1. White bars and black circles denote the median and mean values, respectively. Actual distributions are shown in SM.

to be visually inspected (e.g., Fig 1D–1F). More importantly, our results show that in comparison to same-size threshold and random subgraphs, the metric backbone (of real-World social contact networks and simulated stochastic block models) best preserves community structure (Section 3.1) and is a primary subgraph for simple transmission dynamics (Section 3.2). This suggests that community structure is particularly robust to semi-metric edge removal, and that the observed large amount of shortest-path redundancy shapes epidemic spread in human populations.

Thresholding is often used to decrease the number of edges in densely connected networks. The common assumption is that edges with small proximity (large distance) weights are spurious or simply noise. However, as we show here in all networks studied, the set of weak edges includes those that do in fact contribute to the computation of shortest paths and preserve connectivity and are, thus, important to keep communities connected and sustain spreading phenomena. Thresholding a network can isolate communities, creating disconnected components where dynamical processes, such as epidemic spread, cannot be sustained. Indeed, our epidemic spread simulations on the nine social networks (Section 3.2) show that threshold or random subgraphs much larger than the metric backbone frequently lead to disconnected graphs, but the metric backbone preserves connectivity and shortest paths. The importance of weak edges has long been explored in social networks [17]. Our analysis, thus, suggests that weak edges on the metric backbone are potentially good candidates for interventions that aim at reducing transmission while minimizing reduction of social contact in communities.

It is worth also considering the role of semi-metric edges, those not on the backbone (Section 3.2), as their properties show that the distance backbone methodology is not exclusively a network reduction technique. While all (semi-metric) edges with $s_{ij} > 1$ have null betweenness centrality, the distribution of semi-metric edge distortion (7) typically varies widely, often spanning many scales [13]. Indeed, the distribution of $s_{ij}$ is informative about the robustness of

shortest-paths on a given network. If there are many semi-metric edges with low values of $s$, then the removal of metric edges from the backbone is likely to have a small effect on the distribution of shortest-paths because alternative paths with low distortion are likely to exist. In contrast, if there are high values of $s$, removing some edges on the backbone is likely to result in a large disruption of the shortest-path distribution. Thus, while the size of the backbone captures how robust the distribution of shortest paths is to a random attack on the whole network, the distribution of $s$ captures how robust the backbone itself is to attack.

This knowledge is relevant when devising intervention strategies to disrupt spreading phenomena on social networks. If the backbone is robust, with low central tendency of $s$ and a more homogeneous distribution, then removing edges from it (hindering specific social connections) will have little impact on shortest paths; this is the case of the Exhibit (Ir-Ex) and Workspace (Fr-Wo) networks as shown in Fig 5 (see also Figs P and S in S1 Text). If, on the other hand, the distribution of $s$ is heterogeneous with a large central tendency and variation, then targeting specific edges on the backbone would result in a large disruption to shortest paths. The latter is the case of all school contact networks we have analyzed, especially the French (Fr-HS) and US (US-HS) high school networks as shown in Fig 5 (see also SI for $s$ distributions of all school networks). For instance, the median semi-metric distortion of the US-HS network is $\tilde{s}_{ij} \approx 30$. This means that 50% of the (semi-metric) edges not on the backbone break the triangle inequality by a factor of at least 30. In other words, they refer to social distances that are at least 30 times shorter via indirect paths on the backbone. Thus, deleting edges from this backbone is likely to result in a very large impact on the distribution of shortest paths. Because the backbone is typically very small (for the US-HS $\tau(D) = 7.84\%$), such intervention strategies are easier to design on it than on the whole network, which reveals the importance of targeting the backbone in containing epidemic spread.

The work here presented describes the first application of metric backbones to study epidemic spread on social contact networks. As an initial study, it is limited in scope and can certainly be expanded. For instance, we present results for nine real-World networks and ensembles of synthetic networks. Furthermore, we studied only simple SI spreading dynamics. Understanding the role of shortest-path redundancy in epidemic contact networks will certainly benefit from studying additional networks, especially with more clearly understood and independently measured community structure, as well as more complex spreading models, all of which we leave for future work. Additionally, we have studied social contact networks statically and left considering attack interventions to the network or backbone for future work. Another assumption inherent in the work is that shortest paths—computed as the summation of edge weights along the path—are important. To address that assumption, we intend to include other distance backbones in upcoming studies. Indeed, the metric backbone generalizes to any measure of path length [13], not just the summation of distance edges as in the standard shortest paths of the metric backbone. Thus, other distance backbones based on ultrametric, euclidean, and diffusion distances are likely to be relevant for both understanding the organization of and more complex transmission phenomena on social networks, resulting in novel strategies for analysis and mitigation of epidemic spread.

## 4 Methods

### 4.1 Datasets and network computation

The datasets range in date from 2009 to 2015 and were collected from a number of different social environments, such an elementary school [34], a primary school [45], two high schools [33, 46], a hospital ward [42], a workplace [44], a scientific conference [43], and a museum exhibit [43]. The data were collected using proximity sensors worn by individuals. Most of the

resulting datasets provide a contact record data file, where each row consists of the anonymized IDs of individuals and the time that they met someone, with a time resolution of approximately 20 seconds. Some datasets also contain additional metadata. For instance, in the primary school, class and type (i.e., student or teacher) are provided for each contact ID. Details for each dataset and their analysis are provided Section C in S1 Text.

## 4.2 Module similarity between backbone and original network

Several measures are available to quantify how much the metric backbone is able to preserve the social organization captured by the original network and to compare the value with the social organization captured by null model subgraphs obtained by removing the same number of edges. Given two distance graphs $A(X)$ and $B(X)$ defined on the same set of nodes $X$ from contact or proximity data as described in Section 2, we consider the partitions of $X$ to be the community structure of each graph. Specifically, graph $A(X)$ is partitioned into a set of $m_A$ nonempty modules $A_i$, and graph $B(X)$ is partitioned into a set of $m_B$ nonempty modules $B_j$. We consider strict partitions such that $\cup_{i=1}^{m_A} A_i \equiv X$, $A_k \cap A_l \equiv \emptyset \ \forall k \neq l$, and $A_i \neq \emptyset \ \forall i \in \{1 \ldots m_A\}$. Naturally, $i = 1, \ldots, m_A$ and $j = 1, \ldots, m_B$, with $m_A, m_B \in \mathbb{N}$.

The module partitions we consider are either the metalabels of the original networks or the communities found by community detection algorithms of a target distance graph. The goal is then to compare how similar the module partition of a distance graph is to the partition of its backbone—or threshold or random subgraph of the same size.

We consider various measures for computing similarity between modularity partitions of the same set of nodes, which is an old problem in classification [60]. A straightforward way to obtain a value that captures *bidirectional similarity* of modularity between two distance graphs is to use a normalized measure based on the Jaccard proximity [38] between all pairs of modules with one from each distance graph:

$$y_{AB} = \frac{\sum_{i}^{m_A}\sum_{j}^{m_B} P(A_i, B_j)}{\sqrt{m_A \cdot m_B}} \quad \text{where} \quad P(A_i, B_j) = \frac{|A_i \cap B_j|}{|A_i \cup B_j|} \ . \tag{8}$$

This measures how similar partitions $\{A_1 \cdots A_{m_A}\}$ and $\{B_1 \cdots B_{m_B}\}$ of $X$ are to each other and varies in the unit interval ($y_{AB} \in [0, 1]$). The larger the value, the more similar the partitions.

We also consider a *directional similarity* based on the Jaccard proximity to compare each module in a partition to its most similar counterpart in the other partition ($J_{A \to B}$) and vice-versa ($J_{B \to A}$) as follows:

$$J_{A \to B} = \frac{\sum_{i}^{m_A} \max_{j}^{m_B} P(A_i, B_j)}{m_A} \ , \ J_{B \to A} = \frac{\sum_{j}^{m_B} \max_{i}^{m_A} P(A_i, B_j)}{m_B} \ . \tag{9}$$

These dual measures also vary in the unit interval ($J_{A \to B}, J_{B \to A} \in [0, 1]$). The larger the value of $J_{A \to B}$ the more each module in partition $\{A_1 \cdots A_{m_A}\}$ has a similar counterpart module in partition $\{B_1 \cdots B_{m_B}\}$, and vice versa for $J_{B \to A}$.

A measure of *modularity dispersion*, intuitively, yields a comparison opposite to those provided in Eqs 8 and 9:

$$h_{A \to B} = \frac{\sum_{k}^{m_A} H(\mathcal{A}_i)}{m_A \cdot \log_2(m_B)} \ h_{B \to A} = \frac{\sum_{l}^{m_B} H(\mathcal{B}_j)}{m_B \cdot \log_2(m_A)} \ . \tag{10}$$

$H(\mathcal{A}_i)$ and $H(\mathcal{B}_j)$ denote the Shannon entropy of probability distributions associated with the dispersion of each module of one partition into all modules of the other partition:

$$\mathcal{A}_i = \{S(A_i, B_1), \ldots, S(A_i, B_{m_B})\} \quad \text{and} \quad \mathcal{B}_j = \{S(B_j, A_1), \ldots, S(B_j, A_{m_A})\} \, .$$

$S(A_i, B_j)$ denotes the subsethood [61, 62] of module $A_i$ in module $B_j$, and $S(B_j, A_i)$ denotes the subsethood of module $B_j$ in module $A_i$:

$$S(A_i, B_j) = \frac{|A_i \cap B_j|}{|A_i|} \quad \text{and} \quad S(B_j, A_i) = \frac{|A_i \cap B_j|}{|B_j|} \, .$$

Typically $S(A_i, B_j) \neq S(B_j, A_i)$. Thus, $h_{A \to B} \in [0, 1]$ captures how much the modules of partition $\{A_1 \cdots A_{m_A}\}$ are on average dispersed into the modules of partition $\{B_1 \cdots B_{m_B}\}$, and $h_{B \to A} \in [0, 1]$ the other way around. The smaller the values are, the more the community structure of one network is preserved in the other.

For comparison, in addition to the measures in formulae 8 to 10, we also compute the *Clu-Sim* [63, 64] measure of modularity similarity, which considers module "nestedness," and the *Adjusted Rand Index* [60]. All computed measures are shown in the respective dataset details in Section C in S1 Text.

## 4.3 Threshold and random subgraph generation procedure

We compared the metric backbone to both a threshold and a random subgraph, both of equal size of that obtained with the metric backbone. More specifically, to generate a thresholded subgraph from any network we studied here, we rank and remove the weakest edges (i.e., edges with low proximity or large distance) of the original graph until we reach the same number of edges as in the metric backbone. Similarly, to generate a random subgraph, we simply remove edges at random from the original graph until we reach the same number of edges as in the metric backbone.

## 4.4 Synthetic network generation for stochastic block models

Our synthetic network has 160 nodes and 4 modules (A, B, C, and D), each containing 40 nodes each. The underlying connectivity of the network is given by a predefined Stochastic Block Model (SBM) [51], which takes the size of each module and a module probability matrix that defines the likelihood of nodes in a module to connect to nodes in the other modules. Our module probability matrix has a higher intra-module connection and, in addition, modules C and D have a hierarchical connectivity structure (see Fig 3). Edge weights (i.e., their proximity value, $p_{ij}$, or their isomorphic distance, $d_{ij}$) are sampled from a real contact network, the Socio-Patterns French Primary School (Fr-PS; details in Section C.1). Importantly, edges are sampled taking into consideration if they connect two nodes belonging the *same class* or *across different classes* in the Fr-PS. This provide us a realistic differentiation between *within module* and *across module* connectivity weights. The adjacency matrix of both the original graph and its metric backbone can be seen in Fig 3D. Community detection algorithms and modularity measures are the same as with the contact networks (see Section 4.2). Importantly, to estimate deviations all synthetic network results are based on 10 realizations. Additional 100 iterations were used to compute results for the random subgraphs described in Section 4.3. Additional generation details, and a '*high*' intra-module connectivity synthetic network are described in Section B.

## Supporting information

**S1 Text. Additional method details and results on other contact networks.**
(PDF)

## Acknowledgments

We are thankful to Deborah Rocha for very thorough line editing.

## Author Contributions

**Conceptualization:** Rion Brattig Correia, Alain Barrat, Luis M. Rocha.

**Data curation:** Rion Brattig Correia.

**Formal analysis:** Rion Brattig Correia, Alain Barrat, Luis M. Rocha.

**Funding acquisition:** Alain Barrat, Luis M. Rocha.

**Investigation:** Rion Brattig Correia, Alain Barrat, Luis M. Rocha.

**Methodology:** Rion Brattig Correia, Alain Barrat, Luis M. Rocha.

**Resources:** Alain Barrat.

**Software:** Rion Brattig Correia, Alain Barrat.

**Supervision:** Luis M. Rocha.

**Visualization:** Rion Brattig Correia.

**Writing – original draft:** Rion Brattig Correia, Alain Barrat, Luis M. Rocha.

**Writing – review & editing:** Rion Brattig Correia, Alain Barrat, Luis M. Rocha.

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
