## [Decision Letter · Decision Letter 0]

18 Jul 2022

Dear Dr. Brattig Correia,

Thank you very much for submitting your manuscript "The metric backbone preserves community structure and is a primary transmission subgraph in contact networks" for consideration at PLOS Computational Biology.

As with all papers reviewed by the journal, your manuscript was reviewed by members of the editorial board and by several independent reviewers. In light of the reviews (below this email), we would like to invite the resubmission of a significantly-revised version that takes into account the reviewers' comments.

We cannot make any decision about publication until we have seen the revised manuscript and your response to the reviewers' comments. Your revised manuscript is also likely to be sent to reviewers for further evaluation.

Sincerely,

Feng Fu

Associate Editor

PLOS Computational Biology

Nina Fefferman

Deputy Editor

PLOS Computational Biology

Jason A. Papin

Editor-in-Chief

PLOS Computational Biology

Reviewer's Responses to Questions

**Comments to the Authors:**

Reviewer #1: This paper proposes the metric backbone of contact networks and shows the metric backbone preserves the community structures of the full contact networks and is a primary subgraph in epidemic transmission. I do not recommend that the paper be accepted until the problems raised below are addressed.

Comments

1. The main content of this paper is that the metric backbone can well preserve the community structure of the original network. However, the metric backbone has been defined in the reference 13. I want to know whether the metric backbone defined in this paper is the same as that in the references 13. If not, what is the specific difference? If so, this paper seems to only use numerical simulation to illustrate that the metric backbone can well maintain the community structure of the original network.

2. Figure 2 needs to be redrawn. The function of metric backbone is the focus of this paper, so comparison of modules obtained with metric backbone B(X)(not D(x)), and the threshold subgraph/ random subgraph.

3. What I actually want to know is whether B(X) preserves the community structure of the original network (the adjacency matrix instead of the distance graph D(X)). Similarly, compare the spread similarity of the disease on the adjacency matrix (not D(X)) and the metric backbone.

4. Nine different networks are mentioned in the text, but generally only one network is used as an example when numerical simulations illustrate experimental results. The experimental results are insufficient, and each numerical simulation needs to be performed on at least 4 networks without loss of generality.

5. In Figure 3, it is necessary to select different propagation parameters for experiments.

Minor Comments

1. In Figure 1, the two pictures C and F seem to be just the rearrangement of the two pictures B and E (with gephi). Putting them all together is a bit repetitive, and it's easy to misunderstand that there are two different ways to generate subgraphs. Bur in fact there is only one.

2. Discussion is too long and needs to be shortened.

3. Write out the specific generation method of threshold subgraph.

Reviewer #2: The manuscript entitled as “The metric backbone preserves community structure and is a primary submission subgraph in contact networks” by Correia and others builds on a new network measure, called metric backbone, that was introduced by some of the authors in another publication [13]. This measure allows the identification of the superposition of shortest paths in weighted static networks. This is a very important problem, as its successful solution allows to use a reduced network representation of very dense structures to better explore their hidden structure and use them more efficiently for the modelling of ongoing dynamical processes.

In this contribution, after introduction, the authors apply this new metric on nine real-world networks, which they obtain by aggregating temporal networks recorded in various settings and shared in the SocioPatterns project. As initial results they conclude that the identified backbones radically reduce the number of links while keeping the whole networks connected (e.g. without reducing their node sets). Subsequently they aim to demonstrate how much this method preserves the original (sometimes invisible) community structure of the original graphs as compared to some reference models. I found this part of the manuscript less convincing (a) because they base their claims on the success of one (modularity based) community detection method only (the Louvain method) and (b) although the identified backbone structures outperform the actual reference models, yet they consistently suggest many more community to be present, as compared to the original graph. In the third part of the manuscript they simulate an information spreading model on the constructed distance graph with different levels of reduction and compare their results to the corresponding reference models. Indeed their network outperforms the reference models. They show that even in the identified backbone the size of the reduced edge set is radically smaller, the speed of simulated spreading is not changing proportionally but provide comparable results to the dynamics simulated on the original network. I found this part of the manuscript more convincing.

The paper is well written and scientifically sound. On the other hand, it provides only demonstrative results of an already established methodology. Moreover, although the simulated spreading process can be considered as the simplest example of an epidemic process, the contribution falls rather far from any direct biological application.

Comments:

- The title of the paper is very long and somewhat misleading, as the metric backbone is not precisely preserving the community structure of the original graph but gives a better approximation of it as compared to some reference models.

- Page 3, Section 2, line 1: “Social contact networks are built by recording when individuals meet.”: This is not a precise sentence. Social contact networks are recording with whom people meet. Only their temporal network representation is concerned by the question of “when”. Since this contribution primarily concentrates on static structure, it would be better to rephrase this sentence.

- Section 2: I missed from the introduction of the method the discussion of alternative shortest paths. Which shortest path the method is choosing between nodes if there are two fully equivalent shortest paths between them?

- Section 2: Although the original method is published in another paper, yet I missed some mention about the efficiency of the proposed method. What is the computational and memory complexity? How does the method scale? Can it be applied on very large or very dense weighted networks? This is not demonstrated in the paper as the analysed real networks are rather small.

- Section 3.1: Similar to the the title, I would tone down the title of this subsection.

- As demonstrated in Figure 1, although one would expect that a shortest path sub-graph is more tree like, the metric backbone structure contains surprisingly many triangles. This is probably because it operates with weighted shortest paths, yet it would help the reader if the authors could elaborate on this observation more.

- It would make the contribution stronger if the authors would test with more than one community detection method (using other paradigm as modularity), how much metric backbone “preserves community structure”.

- Table 2: Although the metric backbone outperforms the random and the thresholded networks in terms of identifying similar to real communities, yet the number of the identified communities (17) is significantly larger than originally (10). This is one reason I think it is unfair to put forward as main result that this method “preserves communities”. Actually it breaks them up considerably. The question is whether this is due to the sparsity of the backbone network or due to some bias the algorithm introduces?

- Figure 3: The meaning of the y axis is not well explained in the text or the caption. What “t (partial)” and “t (full network)” means in this context? Are the comparisons fair between networks which have different LCC sizes?

- Figure 3: The authors depict the y axis on a log scale and conclude that the spreading speed “remains much closer” to the original network as compared to the reference models. This is indeed true and it is well demonstrated visually, yet the logarithmic scale hides that there is a factor of 2 difference for the t_1 case and a factor of 4 difference for the t_1/2 cases as compared to D(X). Regarding that the application of this methodology is meant for epidemic modeling, these differences are rather large. Although I find this result appealing and very interesting, this scaling should be discussed in the paper.

- Have the authors considered to compute the metric backbones of other types of networks, e.g. from biology?

- The authors completely missed to discuss the limitations of their results in the Discussion section.

**Have the authors made all data and (if applicable) computational code underlying the findings in their manuscript fully available?**

Reviewer #1: None

Reviewer #2: Yes

PLOS authors have the option to publish the peer review history of their article (what does this mean?). If published, this will include your full peer review and any attached files.

Reviewer #1: No

Reviewer #2: No
---

## [Decision Letter · Decision Letter 1]

6 Jan 2023

Dear Dr. Brattig Correia,

We are pleased to inform you that your manuscript 'Contact networks have small metric backbones that maintain community structure and are primary transmission subgraphs' has been provisionally accepted for publication in PLOS Computational Biology.

Best regards,

Feng Fu

Academic Editor

PLOS Computational Biology

Nina Fefferman

Section Editor

PLOS Computational Biology

Reviewer's Responses to Questions

**Comments to the Authors:**

Reviewer #1: The authors improved the paper and answered all of my questions. Thus, now I suggest publication.

Reviewer #2: I have carefully studied the responses of the authors and read the revised manuscript. The authors addressed all my comments to the expected level, thus I suggest the manuscript for publication.

**Have the authors made all data and (if applicable) computational code underlying the findings in their manuscript fully available?**

Reviewer #1: None

Reviewer #2: Yes

PLOS authors have the option to publish the peer review history of their article (what does this mean?). If published, this will include your full peer review and any attached files.

Reviewer #1: No

Reviewer #2: No

---

## [Editor Report · Acceptance letter]

31 Jan 2023

PCOMPBIOL-D-22-00171R1 

Contact networks have small metric backbones that maintain community structure and are primary transmission subgraphs

Dear Dr Brattig Correia,

I am pleased to inform you that your manuscript has been formally accepted for publication in PLOS Computational Biology. Your manuscript is now with our production department and you will be notified of the publication date in due course.

With kind regards,

Zsofi Zombor
